# Characterization and Biological Activities of Seed Oil Extracted from *Berberis dasystachya* Maxim. by the Supercritical Carbon Dioxide Extraction Method

**DOI:** 10.3390/molecules25081836

**Published:** 2020-04-16

**Authors:** Lijuan Han, Qingqing Han, Yongjing Yang, Honglun Wang, ShuLin Wang, Gang Li

**Affiliations:** 1State Key Laboratory of Plateau Ecology and Agriculture, Qinghai University, Xining 810016, China; hlj880105@163.com (L.H.); yongjing223@163.com (Y.Y.); hlwang@nwipb.cas.cn (H.W.); wangsl1970@163.com (S.W.); 2Center for Mitochondria and Healthy Aging, College of Life Sciences, Yantai University, Yantai 264005, China; ytdxhqq@163.com

**Keywords:** *Berberis dasystachya* Maxim., seed oil, supercritical carbon dioxide, fatty acids, volatile components, antioxidant activity, cytotoxicity

## Abstract

Characterization of the structure and pharmacological activity of *Berberis dasystachya* Maxim., a traditional Tibetan medicinal and edible fruit, has not yet been reported. In this study, central composite design (CCD) combined with response surface methodology (RSM) was applied to optimize the extraction conditions of *B. dasystachya* oil (BDSO) using the supercritical carbon dioxide (SC-CO_2_) extraction method, and the results were compared with those obtained by the petroleum ether extraction (PEE) method. The chemical characteristics of BDSO were analyzed, and its antioxidant activity and in vitro cellular viability were studied by DPPH, ABTS, reducing power assay, and MTT assay. The results showed that the maximum yield of 12.54 ± 0.56 g/100 g was obtained at the optimal extraction conditions, which were: pressure, 25.00 MPa; temperature 59.03 °C; and CO_2_ flow rate, 2.25 SL/min. The Gas chromatography (GC) analysis results showed that BDSO extracted by the SC-CO_2_ method had higher contents of unsaturated fatty acids (85.62%) and polyunsaturated fatty acids (57.90%) than that extracted by the PEE method. The gas chromatography used in conjunction with ion mobility spectrometry (GC–IMS) results showed that the main volatile compounds in BDSO were aldehydes and esters. BDSO also exhibited antioxidant ability in a dose-dependent manner. Moreover, normal and cancer cells incubated with BDSO had survival rates of more than 85%, which indicates that BDSO is not cytotoxic. Based on these results, the BDSO extracted by the SC-CO_2_ method could potentially be used in other applications, e.g., those that involve using berries of *B. dasystachya*.

## 1. Introduction

*Berberis dasystachya* Maxim.is one of the main types of berries from the Qinghai–Tibet plateau [1] and is a traditional Tibetan medicinal and edible plant whose fruit possesses pharmacological properties. In the classical Tibetan pharmacological book known as the “Qinghai–Tibet medicine Kam” (in Chinese, Pinying: Qingzangyaojian; Northwest Institute of Plateau Biology, 1972), the plant is described as having the ability to cure indigestion, ophthalmalgia, and arthralgia. In Qinghai Province, the plant is known as “Yellow-thorn”. However, together with *Nitraria tangurorum* Bobr. (“White-thorn”) and *Hippophae rhamnoides* L. (“Black-thorn”), the plant is collectively referred to as the “Qinghai Three Thorns”. During the 1980s, the three thorn fruits were analyzed and developed to produce “The Cream of Three Thorns” and “the Fruit Juice of Three Thorns” [2]. *B. dasystachya* fruit is rich in bioactive substances such as phytosterols, organic acids, and carotenoids [2,3]. However, previous studies have focused only on *N. tangurorum* and *H. rhamnoides*. Thus, the active phytochemicals of *B. dasystachya* are yet to be identified.

The seeds of many plants have been used as oil sources for nutritional, medicinal, and industrial purposes [4]. However, many of these inexpensive, valuable wild plants remain inadequately utilized and investigated [5]. Fruit or vegetable oils can be used as sources of poly-unsaturated fatty acids for human consumption, but they have potential problems due to bioaccumulation of pesticides, heavy metals, and other hazardous substances. In recent years, plant oils have also been demonstrated to be useful as sources of omega-3 poly-unsaturated fatty acids, which are important nutrients for humans [6,7]. Seed oils from plants like *N. tangurorum* and *H. rhamnoides* are rich in polyunsaturated fatty acids [8,9], which play an important role in physiological and biological functions of living organisms [10,11]. Previous studies have shown that the seed oil of *B. dasystachya* (BDSO) prepared by screw-pressing contains rich antioxidants, with values reported for total polyphenol content (TPC, 281.61 ± 19.06 g/g), total carotenoid content (TCC, 65.43 ± 5.46 µg/g), total steroid content (TSC, 291.74 ± 22.21 µg/g), unsaturated fatty acids (UFAs, 368.61 µg/mL), and polyunsaturated fatty acids (PUFAs, 305.39 µg/mL) [11,12]. Some of these antioxidants could reduce the amount of free radicals produced in vitro. Furthermore, our previous results showed that supplementation of BDSO could decrease the side-effects of fatigue via variation of biochemical parameters and modify relevant antioxidant parameters in order to prevent lipid oxidation in mice exposed to force swimming [12].

Supercritical carbon dioxide extraction (SC-CO_2_) has been used to extract natural products intended for uses in the food and pharmaceutical industries [13]. Compared to the traditional extraction methods using organic solvents, SC-CO_2_ possesses many advantages because it is non-toxic and has non-flammable intermediate products. Moreover, it is efficient as it can produce highly pure oils with high yields, and it meets the growing demands of the market for “natural” products [14]. Methods generally used for oil extraction include expeller pressing and solvent extractions. The SC-CO_2_ method has been successfully employed in the extraction of fatty acids and volatile components from many plant sources [15,16]. Temperature, pressure, extraction time, and CO_2_ flow rate have been optimized to obtain the optimal extraction conditions. Response surface methodology (RSM) has been verified to be a statistical and a mathematical tool for optimizing the reaction parameters, and has also successfully been used to optimize extraction conditions using SC-CO_2_ in pharmaceutical research [17,18]. However, studies on extraction of BDSO using the SC-CO_2_ method have not been reported.

Plants oils are abundant in essential fatty acids and fat-soluble vitamins, while also containing other plant components such as volatile organic compounds (VOCs). Different plant oils have considerably different fatty acid compositions and volatile compounds. Thus, their health efficacy and edible flavor are different [19]. Furthermore, the concentration and composition of VOCs depend on environmental conditions, species, and geographical origin, as well as the drying process and extraction method [20]. Plant oils extracted by different methods contain different volatile compounds, which can interact differently with the surrounding environment. Various studies have reported that many varieties of fruit oils and vegetable oils produced by the SC-CO_2_ fractionation technique contain many high-quality volatile compounds [21,22].

The technique currently and widely used for analysis of volatile compounds in plant oils is gas chromatography coupled to mass spectrometry (GC–MS) [23]. However, GC–MS is only limited to laboratory use and requires helium gas supplies. Thus, it is not suitable for inline analysis or small laboratories [24]. Another technique that allows the authentication of VOCs during the analysis of adulteration of plant oils is based on ion mobility spectrometry (IMS). The technique has high sensitivity, and can detect analytes at very low concentrations in the range of 0.01–0.05 ppm, which is usually below the odor threshold of many major volatiles [25]. Thus, gas chromatography used in conjunction with ion mobility spectrometry (GC–IMS) is an analytical technique that can detect trace gases. The instrument can also allow the mobility spectra to be automatically acquired upon chromatographic elution of each target compound. Thus, several spectra at a given retention time can be simultaneously processed to obtain information on both retention time and drift time, and larger amounts of analytical information from each sample can be obtained [26,27]. GC–IMS can not only detect compounds from different plant origins, but can also be used for evaluating and characterizing various products and vegetable oils [24,26,28]. At present, however, information on the changes of volatile components in oils of preserved *B. dasystachya* seeds during extraction processing is unavailable.

This work studied variations of flavors of preserved *B. dasystachya* seeds during extraction to infer the mechanism of flavor changes by monitoring different flavor compositions generated during different extraction methods. Flavor changes can also be used as criteria for the differences in the quality of preserved BDSO. In this way, the operation or extraction conditions can be optimized to produce preserved BDSO with tastes that meet the different preferences of consumers. Moreover, there are limited data information currently available on the phytochemical composition and bioactivity of seed oils extracted from *B. dasystachya*. Thus, the objective of this study is to optimize the SC-CO_2_ process of *B. dasystachya* seed oils by comparing the products obtained by the petroleum ether extraction (PEE) and SC-CO_2_ extraction methods. To obtain an oil product with unique quality and to achieve maximum oil yield, the conditions for SC-CO_2_ extraction were optimized using a five-level, three-variable central composite design (CCD) based on RSM. The SC-CO_2_ extraction conditions, including pressure, temperature, and CO_2_ flow rate, were optimized to obtain the highest oil yield. The phytochemical composition of the oil extracted by the SC-CO_2_ method was evaluated by Fourier-transform infrared spectroscopy (FT-IR), thermogravimetric analysis (TG), differential scanning calorimetry (DSC), gas chromatography with flame ionization detector (GC–FID), and gas chromatography used in conjunction with ion mobility spectrometry (GC–IMS), and was compared with that of the oil extracted by the PEE method. The antioxidant activities of the extracted oil against three types of free radicals of the extracted oil were determined by scavenging assays, and the in vitro cytotoxicity of the oil was determined by MTT assay.

## 2. Results and Discussion

### 2.1. Single-Factor Experiments

Figure 1a–c shows the effects of extraction pressure, extraction temperature, and CO_2_ flow rate on the oil yield of BDSO. Extraction pressure is an important extraction parameter that can significantly affect the efficiency of the SC-CO_2_ extraction method [17]. Pressure of above 35 MPa can not only lead to high energy consumption, but can also reduce the safety of the equipment. In this experiment, the extraction pressure was varied (10, 15, 20, 25, and 30 MPa), while other parameters including extraction temperature and CO_2_ flow rate were set as constant values at 50 °C and 2.0 SL/min, respectively. As shown in Figure 1a, the yield of BDSO increased as the pressure was increased from 15 to 25 MPa. Additionally, the extraction yield increased from 12.11 ± 1.32%, reaching its maximum value of 12.45 ± 1.65% at 20 MPa. After that, it increased slightly as the increasing extraction pressure continued to rise. Therefore, it can be concluded that 20 MPa is the optimal extraction pressure.

Temperature is another factor that can significantly influence the efficiency of the SC-CO_2_ extraction process [29]. The extraction process was carried out at varying temperatures of 30, 40, 50, 60, and 70 °C, while the extraction pressure was set at 20 MPa and the CO_2_ flow rate was set at 2.0 SL/min. Figure 1b shows that the extraction yield increased up to a maximum value of 12.42 ± 1.87% at a temperature of 60 °C. The BDSO yield was not increased when the extraction temperature was raised to 70 °C, indicating that higher extraction temperatures may induce oil degradation, thus causing the yield to reduce.

CO_2_ flow rate is also an important factor that can influence the efficiency of the SC-CO_2_ extraction process [30]. The CO_2_ flow rate can affect the extraction yield of seed oil in two ways as follows: it causes the oil yield to increase, and it can decrease the contact time between the fluids and the materials, thus leading to low yield. For this experiment, the CO_2_ flow rates were set at 1.0, 1.5, 2.0, 2.5, and 3.0 SL/min, while the extraction pressure and temperature were set at 20 MPa and 50 °C, respectively. The results depicted in Figure 1c shows that when the CO_2_ flow rate was increased from 1.5 to 2.5 SL/min, the extraction yield increased, reaching its maximum of 12.46 ± 1.23% at 2.5 SL/min, and decreased thereafter. These results indicate that the optimum CO_2_ flow rates were in a range of 1.5 to 2.5 SL/min.

### 2.2. Optimal Extraction Conditions Obtained by Central Composite Design

#### 2.2.1. Optimal SC-CO_2_ Conditions

The BDSO yields obtained from the CCD experiment are shown in Table 1. Based on the coefficients determined by the *t*-test, the effects of temperature and extraction time on the BDSO yield were significant at *p* < 0.001, and that of CO_2_ flow rate was also significant but at a lower level of *p* < 0.05. These results indicated that extraction pressure and temperature were the major factors contributing to the BDSO yield.

Central composite design (CCD) was used to evaluate the effect of SC-CO_2_ parameters on the BDSO yield. The second-order polynomial regression model, which shows the experiential relationship and the interactions between dependent (*Y* = BDSO yield) and independent variables (*X_1_*= pressure, *X_2_*= temperature, *X_3_*= CO_2_ flow rate), is described in Equation (1). Table 2 shows the multiple regression coefficients and the predicted second-order polynomial model obtained by the least-squares technique for the oil yield, which indicate that the quadratic parameters were significant (*p* < 0.01). The regression model for BDSO yield was highly significant (*p* < 0.01). The R^2^ value (0.9768), which can evaluate the adequacy of the model, indicated that the model reasonably fitted with the experimental data. The adjusted R^2^ value (0.9560) and the predicted R^2^ value (0.8679) also confirmed that the experimental data were in agreement with the predicted data. Finally, the F-value for lack of fit (calculated by ANOVA) was not significant relative to the pure error, implying that our model could sufficiently accurately predict the responses. The coefficient of variation (CV) was 1.38%, confirming that our model has high precision and reliability.
(1)$$Y=12.32+0.30X_{1}+0.32X_{2}+0.22X_{3}+0.21X_{1}X_{2}-0.099X_{1}X_{3}+0.15X_{2}\;X_{3}-0.35X_{1}^{2}-\;0.25X_{2}^{2}-0.57\;X_{3}^{2}$$

#### 2.2.2. Validation of Models

The data were evaluated by ANOVA, and the effects of the variables on the extraction efficiency predicated by the second-order polynomial model are summarized in Table 3. The linear and quadratic values of pressure (*X_1_*), temperature (*X_2_*), and the CO_2_ flow rate (*X_3_*) were highly significant (*p* < 0.01). The interaction between extraction pressure and temperature was highly significant (*p* < 0.01) in terms of BDSO yield. Similarity, the interaction between extraction temperature and CO_2_ flow rate was statistically significant (*p* < 0.01). The interaction between extraction pressure and CO_2_ flow rate within the experimental range was also significant (*p* < 0.05). Thus, the analysis of the linear and quadratic values of the parameters and their interactions showed that the variables analyzed in this study were the most important factors impacting the BDSO yield.

We constructed three-dimensional shaded surface and contour plots by which one variable was kept constant at the center value of the experimental ranges (zero level) at a certain time, while the other two variables were varied. These plots were used to predict the interaction between the experimental parameters. While the three-dimensional presentation of the impact of the variables can help us to find the maximum, the minimum, and the saddle points of the responses, the two-dimensional contour plots can indicate the levels of the variables in a curve that have equal responses, thus allowing an easy interpretation of the optimal values of the variables that lead to the maximum yield.

Figure 2 illustrates the three-dimensional response surface (Figure 2a) and contour plots (Figure 2b) of the effects of pressure and temperature on the BDSO yield. The oil yield increased with the increase in both the extraction pressure and the extraction temperature. The maximum BDSO yield was achieved when the extraction pressure and temperature were 24.07 MPa and 59.50 °C, respectively. With increasing extraction temperature, the BDSO yield increased until it reached a plateau, and slightly declined thereafter.

Figure 3 depicts the effects of pressure and CO_2_ flow rate on the BDSO yield at a constant extraction temperature of 50 °C. According to the curve, at a given temperature, the BDSO yield increased with increasing pressure. At low CO_2_ flow rates in particular, the BDSO yield increased linearly with increasing pressure. However, at CO_2_ flow rates of higher than 1.92 SL/min, the BDSO yield increased only slightly with increasing pressure, indicating that the effect of pressure on the oil yield at these CO_2_ flow rates was weak. The BDSO yield reached its maximum value at the extraction pressure and CO_2_ flow rate of 21.92 MPa and 1.92 SL/min, respectively. Thus, we may conclude that the effect of pressure on the extraction oil yield was more prominent at low CO_2_ flow rates compared to that at high CO_2_ flow rates.

Figure 4 shows the three-dimensional response surface and contour plots for the effects of temperature and CO_2_ flow rate on the oil yield. The oil yield increased with increasing temperature, indicating that temperature can influence the yield of oil extracted from *B. dasystachya* seeds, which is likely because higher temperature can increase the solubility of lipid. However, at temperatures of higher than 48.30 °C, the oil yield decreased. We hypothesize that there may a link between temperature and density of fluid, or it is possible that high temperature can break the balance between fluid and pressure [18,30].

#### 2.2.3. Model Analysis

Furthermore, the optimal values of the three variables obtained from the Design–Expert software were 25.00 MPa, 59.03 °C, and 2.25 SL/min (Table 4). The BDSO yield averaged from four validation experiments carried out under the optimal conditions with some slight changes was 12.54 ± 0.56 g/100 g (*w*/*w*, *n* = 4), which is not significantly different from the predicted oil yield (12.553 g/100 g). This result further proved that regression model can adequately predict the optimal extraction conditions in the laboratory.

### 2.3. Seed Oil Yield

The yield of BDSO extracted by the petroleum ether extraction (PEE) method was 10.76%. The BDSO yields obtained by SC-CO_2_ extraction at different conditions are present in Table 3. Previous studies [12] have shown that the yield of seed oil obtained by screw-pressing was 5.32 ± 1.72%, which was lower than the BDSO yield extracted by PEE and SC-CO_2_. The yield of BDSO extracted by the SC-CO_2_ method ranged from 10.39% to 12.44%, which was higher than that extracted by the PEE method, suggesting that the extraction of BDSO by the SC-CO_2_ method is superior to that by the PEE method. This is in agreement with previously reported results [17]. In addition, it is worth noting that BDSO extracted by the PEE method contained traces of organic solvents. The solvent residue left in the oil is considered to be toxic to humans and animals. Therefore, other operational steps such as evaporation concentration and nitrogen concentration, etc., are needed after the extraction is completed to obtain the required samples. These steps will lead to the loss of oil yield. SC-CO_2_ extraction is efficient, safe, and can reduce the use of organic solvents for oil extraction. The SC-CO_2_ method could extract BDSO with a superior extraction yield compared to conventional PEE methods, as described previously [17,31]. Thus, we can conclude that SC-CO_2_ extraction method is an acceptable and environmentally friendly procedure for oil extraction from *B. dasystachya* seeds.

### 2.4. FT-IR Spectra

Figure 5 shows the Fourier-transform infrared (FT-IR) spectrum of BDSO. Various functional groups present in the BDSO obtained by different methods were identified from the FT-IR analysis, as shown in Table 5. The absorption bands of BDSO at 2348.87 and 2327.65 cm^−1^ (Figure 5a) were assigned to bending vibration of C≡ stretch, which clearly indicates that BDSO contains CO_2_. This is in agreement with previously reported results [32]. The absorption bands of BDSO at 1500.34 cm^−1^ (Figure 5a) were assigned to a C=C stretch on the benzene ring, which indicates that BDSO obtained by SC-CO_2_ contains benzenoid hydrocarbon. The absorption bands of BDSO at 1465.17 cm^−1^ (Figure 5a), 1461.77 cm^−1^ (Figure 5b), and 1745 cm^−1^ were assigned to bending vibration of lipid CH_2_ groups and the ester carbonyl stretching (C=O) of fatty acids, respectively [32]. The FT-IR spectrum clearly indicates that BDSO contains =C-H, C=C, and -C=C=C- groups, suggesting that the oil is an unsaturated fatty acid containing ester groups [33]. The absorption bands at 1745 and 900~1200 cm^−1^ indicate the presence of ester groups, suggesting that the oil obtained from different extraction methods can be transformed into other types of ester, which can serve as a low-grade feedstock for biodiesel synthesis. However, compared with the previous research results [12], the infrared spectra of the three kinds of seed oil (BDSO obtained by screw-pressing, PEE, and SC-CO_2_, respectively) have similar characteristic absorbance bands such as hydroxyl groups (around 3000 cm^−1^), -CH_2_ (around 1460 cm^−1^), -CH_3_ (around 1377 cm^−1^), and -(CH_2_)_n_- (*n* ≥ 4, around 720 cm^−1^).

### 2.5. Fatty Acid Composition

Table 5 presents the fatty-acid compositions of BDSO extracted by PEE and the SC-CO_2_ extraction method. The fatty acids in BDSO extracted by the two methods mainly included saturated fatty acids (SFAs) such as undecanoic acid (C11), tridecanoic acid (C13), myristic acid (C14), pentadecanoic acid (C15), palmitic acid (C16), stearic acid (C18), and eicosanoic acid (C20), and unsaturated fatty acids (UFAs) such as palmitoleic acid (C16:1), oleic acid (C18:1), linoleic acid (C18:2), linolenic acid (C18:3), and erucic acid (C22:1). In the previous study [12], the fatty acid content of BDSO obtained by screw-pressing was analyzed by GC–MS, and the result was different from that of GC analyses in this paper: six saturated fatty acids (lauric acid, myristic acid, pentadecanoic acid, heptadecylic acid, stearic acid) and six unsaturated fatty acids (palmitoleic acid, oleic acid, linoleic acid, γ-linolenic acid, α-linolenic acid, eicosenoic acid) were identified. Furthermore, it was found that linolenic acid had the highest unsaturated fatty acid content in BDSO obtained by both PEE (32.28 ± 2.02%) and SC-CO_2_ extraction (34.74 ± 1.91%), which was consistent with the results of screw-pressed seed oil (linolenic acid: 170.59 ± 9.63 µg/mL) in the previous study [12]. These results indicate that the seed oil from *B. dasystachya* is a good source of linolenic acid.

Fatty acid content in BDSO extracted by the PEE method was significantly different from that extracted by the SC-CO_2_ method. The UFAs obtained by the SC-CO_2_ method accounted for 85.62% of the total fatty acids. The composition and the ratio of saturated/unsaturated fatty acids (SFAs/UFAs) are indicators that can be used to evaluate nutritional and functional characteristics of oils. The SFA/UFA ratios of BDSO obtained from the PEE and the SC-CO_2_ methods were 0.2364 and 0.1678, respectively. BDSO obtained by the SC-CO_2_ method was rich in both UFA (nearly 85.62% of the total fatty acids) and polyunsaturated fatty acids (PUFA; nearly 57.90% of the total fatty acids). Both the UFA and PUFA contents obtained in the oil extracted by the PEE method were lower than those in oil extracted by the SC-CO_2_ method. According to the references, there may be three reasons for this result. Firstly, the SC-CO_2_ extraction method has a superior BDSO yield compared with that extracted by the conventional PEE methods. Secondly, in the extraction of fatty acids using an organic solvent, the samples are concentrated using a rotary evaporator at high temperature before direct injection, and are further reduced under a stream of N_2_ [34]. These concentration steps require the use of high temperatures, which leads to the thermal degradation of bioactive compounds such as polyunsaturated fatty acids [35]. By contrast, in the extraction of fatty acids by the SC-CO_2_ method, the oil is transferred directly into the injection port of the GC or GC–MS system for analysis [36], and the extraction process is carried out at low temperatures, which can minimize thermal damage to UFAs and PUFAs in the oil. Furthermore, carbon dioxide reaches the critical point under relatively mild conditions, and extraction in the absence of oxygen is beneficial for the preservation of bioactive compounds (PUFAs, UFAs) without oxidation [37]. Based on the above, we can conclude that SC-CO_2_ extraction can extract BDSO with superior quality and higher concentration of polyunsaturated fatty acids compared to the conventional PEE methods.

### 2.6. Volatile Organic Compounds

The volatile organic compounds (VOCs) of BDSO with different processing methodologies (i.e., PEE and SC-CO_2_) were determined via GC–IMS analysis. A Flavor Spec^®^ instrument was used to generate the three-dimensional spectra of the data, which can show the difference of the matrix plot of the average GC–IMS spectra (Figure 6). The three-dimensional spectra of VOCs were highly complex, presenting more than 79 individual signals. The VOCs in different samples had varying peak intensities. The two-dimensional spectra of these individual identifiers (features) obtained at the time of the measurement and the normalized drift time are shown in Figure 7a. The red region indicates that the oils obtained by the PEE method contained higher contents of VOCs compared with those obtained by the SC-CO_2_ extraction method, particularly for those with identifier numbers 20–21, 23, 25–26, 28, 31–32, and 34, which were not detected in BDSO obtained by the SC-CO_2_ method, but were detected at high signals in BDSO obtained by the PEE method. The higher the intensity of the red color, the higher the concentration of VOCs contained in the oils; the blue region has the opposite interpretation. Figure 7a shows that the highest concentration of some VOCs appeared at run times between 50 s and 200 s. These signals are typical of VOCs (e.g., acids, alkanes, aldehydes, and alcohols) found in edible oils [38,39]. However, the intensities of the signals of these compounds were varied slightly.

An area set that integrates all the marker peaks was created to allow the data to be viewed more intuitively (Figure 7b). The spectra of BDSO obtained by the PEE method contained a greater number of peaks than that obtained by the SC-CO_2_ method, and the spectral profiles were also largely different. This clearly indicates that the oils obtained by different methods could be distinguished using a non-targeted profiling approach without the need for chemical markers or calibration curves. The intensities of some characteristic signals attributed to the presence of the characteristic VOCs also varied. The numbers of VOCs in BDSO obtained by the SC-CO_2_ method showed a decreasing trend, and their concentrations were also lower compared with those of VOCs obtained by the PEE method. This result shows some correlations between the extraction methods and the trends of VOCs.

To identify the characteristic VOCs in the topographic GC–MS plot of BDSO samples, VOCs representing different chemical classes were studied by a user-defined VOC area set library created using the LAV software. Table 6 shows the VOCs of BDSO, and Figure 7 shows the qualitative analysis of each type of flavor compounds with the same serial numbers presented in Table 6. Based on the fingerprints, 78 peaks corresponding to 35 compounds, as well as their quality, were observed in all types of BDSO. These compounds included 7 esters, 17 aldehydes, four ketones, 5 alcohols, 1 terpene, and 1 organic acid. The highest number of compounds (33) was found in BDSO extracted by the PEE method, whereas the lowest number of compounds (26) was found in BDSO extracted by the SC-CO_2_ method. This result is inconsistent with other reports focused on the volatile compounds of rattan pepper oil obtained by PEE and SC-CO_2_ [40]. This difference may be attributed to different fruit matrices, genetics, nutrition, and growth environment [41]. On the contrary, a new VOC (isoamyl acetate) was also found in the oil obtained by the SC-CO_2_ method, accompanied by the disappearance of some other substances. The concentrations of some volatiles, including ethyl lactate, octanal, benzaldehyde, furfural, 2-methyl-propanal, cyclohexanone, and 2-butoxyethanol, in BDSO obtained by the PEE method were higher than those obtained by the SC-CO_2_ method (peak intensity ≥ 500). The substances in the red rectangle (Figure 7b) could be considered as providing the characteristic flavor compounds in BDSO obtained by the PEE method. A total of 24 compounds were observed in both types of BDSO, among which three compounds including hexanal, acetone, and ethanol were the main compounds (peak intensity ≥ 1500). Most of these compounds are responsible for the fruity/cut-grass aroma of oils and were found at high levels in both types of BDSO. Aldehydes and esters were also found to be the main VOCs in both types of BDSO; however, their contents were relatively different.

Typically, aldehydes are considered as important compounds that influence the aroma of plant oils due to their relative low thresholds and higher concentrations [42,43]. In general, aldehydes result from the endergonic metabolic processes of fatty acids and the decarboxylation of amino acids [43,44]. The results showed that the contents of two compounds (heptanal and pentanal) in BDSO obtained by the PEE method were higher than those obtained by SC-CO_2_ method, which may be because unsaturated fatty acids can be oxidized by air during air-extraction. This finding is consistent with a report by Javed et al. [45], which demonstrated that heptanal and pentanal compounds in sun-dried and air-dried raisins are derived from unsaturated fatty acid oxidation. The increases in the intensities of benzaldehyde and 3-methylbutanal in BDSO extracted by the PEE method were much higher than those in BDSO extracted by the SC-CO_2_ method, and the increases were more accentuated with the increase in temperature. This observation could be explained by the Strecker degradation of isoleucine, which produces benzaldehyde and 3-methylbutanal during the Maillard reaction in BDSO obtained by PEE [44,46]. The signal intensity of n-nonanal in BDSO obtained by the PEE method was higher than that in BDSO obtained by the SC-CO_2_ method; a similar trend was also observed in benzaldehyde. This may be caused by the thermal degradation of sugars, such as fructose and glucose, which can produce furan-containing compounds, e.g., furfurals [47].

The normalized signal intensity of esters in BDSO extracted by the SC-CO_2_ method increased markedly, which is consistent with the observation by Hu et al. [46]. The increment of ethyl acetate content in BDSO extracted by the SC-CO_2_ method was more accentuated that that in BDSO extracted by the PEE method, and the content of ethyl acetate in the former was approximately 2.2 times higher than that in the latter. Talens et al. [48] have described how due to the absence of gas (oxygen) in the intercellular space during the extraction process, the use of vacuum pulse osmosis can promote ester formation in kiwifruit; thus, low oxygen caused by the presence of CO_2_ during the SC-CO_2_ extraction may promote the production of ethyl acetate in fruits. In addition, the intensity of terpenes (α-pinene) in BDSO obtained by the PEE method was higher than that in BDSO obtained by the SC-CO_2_ method; the percentage of α-pinene in the former was 13.6 times higher than that in the latter. In general, the content of terpenes, particularly for α-pinene, tends to increase after solvent processing, which can be possibly be explained by thermal degradation and metabolism of carbohydrates and lipids [49].

The overall profile of VOCs in BDSO obtained by the PEE method was more distinct compared to that in BDSO obtained by the SC-CO_2_ method, which may be attributed to the conclusion that the conventional extraction method could enrich several volatile substances. The presence or absence of specific VOCs in BDSO may vary depending on external factors such as extraction temperature, extraction time, extraction agent, or oxygen content (for extraction that is exposed to air). In future work, it is necessary to use quantitative methods or GC–TOF–MS to determine more volatile compounds, and to more comprehensively explore the effect of the two extraction approaches on the flavors of the oils.

### 2.7. Thermal Stability and Thermal Behavior

The thermogravimetric (TG) and DSC curves of BDSO obtained by the PEE and SC-CO_2_ methods are shown in Figure 8. According to the first region of the TG curves, masses within the plateau did not change until decomposition occurred. Then, the initial decomposition temperatures of BDSO obtained by the PEE and SC-CO_2_ methods were observed at approximately 358.6 °C and 384.2 °C, respectively. The difference in initial decomposition temperatures of both types of BDSO may be due to the difference between fatty acid chain length, degree of unsaturation, and branching, which are the parameters that affect the thermal stability of oils [50]. The continuous, smooth, steep descending line in the TG curve further indicated that there was a rapid loss of mass, of which the rate can vary with the rate of thermal decomposition of polyunsaturated, monounsaturated, and saturated fatty acids. The temperature at which 98.05% mass loss of BDSO extracted by the SC-CO_2_ method occurred was 587.6 °C, whereas the temperature at which 96.44% mass loss of BDSO extracted by the PEE method took place was at approximately 597.6 °C. The rate of mass loss of BDSO extracted by the SC-CO_2_ method was higher than that of BDSO extracted by the PEE method, which may be because the unsaturated fatty acids in the former are less stable than saturated fatty acid in the latter, thus causing them to become more vulnerable to high temperature.

The DSC curves of extracted oils can illustrate their glass transition temperature accompanied by their crystallization and melting behaviors, which are the parameters generally used to analyze the thermal behaviors of oil samples [33,51]. The DSC curves (at temperatures from 0 °C to 700 °C) of the BDSO are illustrated in Figure 8. The onset temperature is the temperature at which the sample begins to decompose. The BDSO extracted by the PEE method had a higher initial decomposition temperature (115.2 °C) than that extracted by the SC-CO_2_ method (88.9 °C). The highest exothermic peaks of BDSO extracted by the SC-CO_2_ method was observed in the DTG curve at approximately 373.6 °C, which may be caused by distillation, oxidation reactions, and bond scission. At high temperatures, oxygen molecules can bind to hydrocarbon molecules to produce various oxygenated substances that can offset the distillation and evaporation effects, thus causing the reduction of oil mass and the loss of diversity in air [51]. The BDSO extracted by the SC-CO_2_ method had a higher end exothermic temperature (456.1 °C) than that extracted by the PEE method (422.3 °C). Overall, the BDSO extracted by the SC-CO_2_ method was more thermostable than that extracted by the PEE method.

### 2.8. In Vitro Antioxidant Ability

#### 2.8.1. DPPH Radical Scavenging Activity

Using DPPH reagent is an effective method for investigating free radical scavenging activities of various samples. DPPH is a stable free radical that shows a maximum absorption at 517 nm, and in the presence of antioxidants, its color can change from purple to yellow. As shown in Figure 9a, BDSO possessed DPPH radical scavenging activities in a dose-dependent manner, likely because the BDSO can act as a hydrogen donor to that can convert DPPH to DPPH-H. At concentrations ranging from 0.09 to 12.00 mg/mL, the DPPH radical scavenging activity values of BDSO extracted by the SC-CO_2_ method were between 38.24% and 89.56%, and were significantly higher than that of BDSO extracted by the PEE method.

#### 2.8.2. ABTS Radical-Scavenging Activity

We also monitored the ABTS radical-scavenging activity of BDSO via an absorbance of ABTS+ at 734 nm. In the presence of BDSO at various concentrations, the absorbance of ABTS+ decreased in a dose-dependent manner. As shown in Figure 9b, BDSO at different concentrations exhibited different ABTS radical-scavenging activities. The ABTS scavenging ability values of BDSO extracted by the SC-CO_2_ method ranged from 79.89% to 92.60% when the concentrations were varied from 0.09 to 12.00 mg/mL. Moreover, the scavenging ability value of BDSO extracted by the PEE method was lower than that extracted by the SC-CO_2_ method.

#### 2.8.3. Reducing Power Assay

Increasing absorbance at 700 nm can indicate an increase in reducing power. As illustrated in Figure 9c, which shows the reducing power of BDSO at various concentrations, the reducing power of BDSO extracted by the SC-CO_2_ method increased (from 0.1425 to 0.828) with increasing BDSO concentration, indicating that the reducing power of BDSO is concentration-dependent. The results further showed that the BDSO extracted by the SC-CO_2_ method is a better source of antioxidants, as indicated by its higher reducing power, than that extracted by the PEE method.

Compared with previous study [12], the seed oils of *B. dasystachya* prepared by three methods (screw-pressing, PEE, and SC-CO_2_) exhibited prominent DPPH, ABTS radical-scavenging, and Fe^3+^ reducing activities in chemical-based assays. The activities of the seed oils showed a positive correlation to the individual concentrations. In comparison with BDSO obtained by PEE and screw-pressing, highly concentrated (12 mg/mL) seed oils obtained by SC-CO_2_ could scavenge 89.56% and 92.60% of DPPH and ABTS radicals, respectively. The scavenging activity of the three seed oils (screw-pressing, PEE, and SC-CO_2_) upon addition of DPPH radical and the reducing power followed the order: SC-CO_2_ > PEE > screw-pressing. The scavenging activity of the three seed oils upon addition of ABTS radical followed the order: SC-CO_2_ > screw-pressing > PEE. Studies have revealed that the potential antioxidant function of plant oils is mainly because of polyunsaturated fatty acids (PUFAs), tocopherols, and other components [52]. Therefore, this sharp contrast might be due to the composition as well as the concentration of any anti-oxidant compounds present in the seed oils. At this point, BDSO extracted from SC-CO_2_ method could be considered as a valuable raw material for designing natural antioxidants.

### 2.9. In Vitro Cytotoxicity

The effects of BDSO extracted by the SC-CO_2_ and PEE methods on the cell viability are reported in Figure 10. Figure 10a,c showed the effects on viability of tumor cells and normal cells, respectively, of BDSO extracted by the SC-CO_2_ method. Figure 10b,d showed those of BDSO extracted by the PEE method. According to the data, at a concentration range of 1–100 μg/mL, neither sample had obvious cytotoxicity. At a concentration as high as 200 μg/mL for both samples, the cell viability remained higher than 85%, indicating that the BDSO extracted by the two methods is non-toxic to both normal cells and cancer cells.

## 3. Materials and Methods

### 3.1. Materials and Chemicals

The seeds of *B. dasystachya* were collected from healthy and ripe fruits harvested in Huzhu County, Qinghai Province, China. The seed content of the dried fruits was determined as 41.13%. The separated seeds were dried at 60 °C for 36 h in a forced-air oven. The moisture content of the seeds was 6.29 ± 0.013%. The dried seeds were ground into powder. The particles retained on a 40-mesh sieve were collected and then stored at 4 °C for further use. Standards of all fatty acids were of chromatographic grade and purchased from Sigma Reagent Co. (St. Louis, MO, USA). 2,2′-hydrazine-bis(3-ethyl-benzo-thiazoline-6-sulfonic acid) diamine (ABTS), 2,2-ddiphenyl-1-picrylhydrazyl (DPPH), and 3-(4,5-dimethylthiazol-2yl)-2,5-diphenyltetrazolium bromide (MTT) were purchased from Sigma Reagent Co. (St. Louis, MO, USA). All other chemicals and solvents were of analytical grade and used without purification. Three types of cancer cells (human HGC-27 gastric cancer cells, human SH-SY5Y neuroblastoma cells, and rat C6 glioma cells) and three types of normal cells (mouse 3T3-L1 preadipocytes, rat H9C2 cardiomyoblast cells, and human umbilical vein endothelial cells) were used in this study. All these cells were purchased from Culture Collection of the Chinese Academy of Sciences, Shanghai, China.

### 3.2. Petroleum Ether Extraction

Solvent extraction of BDSO was conducted by using the method of Wang et al. with some modifications [4]. The seeds of *B. dasystachya* (20 g) were mixed with 400 mL of petroleum ether (60–90 °C) in a Soxhlet extractor and extracted at 80 °C for 7 h. The solvent was evaporated with a rotary evaporator at 55 °C and flushed with nitrogen. The seed oil was weighed to calculate the yield. Then, the seed oil was placed into a brown glass vial, and stored at −20 °C until it was used to analyze the chemical composition and the biological activities.

### 3.3. Supercritical Fluid Extraction with Carbon Dioxide

The SC-CO_2_ equipment (0.5 L capacity) used was model TC-SFE-50-0.5-120S (Shenyang Suofu’er Co. Ltd., Shenyang, China). Each extraction was performed using 200 g of ground *B. dasystachya* seeds. The samples were placed in a thick-walled stainless-steel extractor with filter plates to ensure that the plant material remained at the bottom of the extractor. Liquefied carbon dioxide cooled through a glycol chiller was pumped into the extraction vessel via a high-pressure pump to the desired pressure. The vessel temperature was regulated by a surrounding heating jacket. The accuracy of pressure and temperature was regulated to ±0.5 MPa and ±1.0 °C, respectively. The flow rate of CO_2_ in the system was controlled by back-pressure regulating valves.

### 3.4. Determination of Extraction Yield

The yield of BDSO was calculated by Equation (2):(2)$$\mathrm{Oil}\;\mathrm{extraction}\;\mathrm{yield}\;={\;W}_{oil}/W_{sample}$$
where W*_oil_* is the mass of BDSO obtained from the sample, and W*_sample_* is the mass of the sample before extraction. The extraction yields are expressed as mean values ± standard deviations.

### 3.5. Experimental Design and Statistical Analysis

A single-factor test was adopted to determine the preliminary range of the extraction variables at first [53]. The extraction variables for optimization including pressure (MPa, *X*_1_), temperature (°C, *X*_2_), and CO_2_ flow rate (SL/min, *X*_3_) were determined after the single-factor test. A five-level-three-variable CCD was applied to optimize the processing parameters for SC-CO_2_ production of BDSO (Design-Expert 7.1.3 Trial, State-Ease, Inc., Minneapolis, MN, USA) [51]. In Table 7, these three factors are arranged into five levels, coded as +1.682, +1, 0, −1, and −1.682.

During extraction, the extraction time was set as 50 min. The pressure, temperature, and CO_2_ flow rate ranged between 11.59 and 28.41 MPa, 33.18 and 66.32 °C, and 1.16 and 2.84 SL/min, respectively. The extraction yield was obtained by weighing the solvent-free oil which was flushed with nitrogen. All experiments were carried out three times. The yield of seed oil (%) was calculated using Equation (2).

In Equation (3), Y is the dependent variable (yield of seed oil), k is the number of variables, β_0_ is the constant term, β_j_, β_jj_, and βi_j_ represent the coefficient of the first-order terms, quadratic terms, and interaction terms, respectively, and *X*_i_ and *X*_j_ are the independent coded variables [54].The statistical significance and the significance of the regression coefficients were confirmed by F-test at a probability (*p*) of 0.05 or 0.01 and evaluated by ANOVA. Values of *p* < 0.05 were regarded as significant.
(3)$$Y=\beta_{0}+\sum_{j=1}^{k}\beta_{j}X_{j}+\sum_{j=1}^{k}\beta_{jj}X_{j}^{2}+\sum \sum_{i<j}^{k}\beta_{jj}X_{i}X_{j}\left(k=3\right)$$

### 3.6. Chemical Characteristics

#### 3.6.1. FT-IR Analysis

FT-IR spectrometer (Bruker Tensor 27, Bruker Corporation, Billerica, MA, USA) was used for the FT-IR spectra acquisition. The sample was mixed with spectroscopic grade potassium bromide powder, then ground and pressed into a 1-mm pellets. FTIR spectrum of the pellet sample was measured in the frequency range of 4000 cm^−1^–400 cm^−1^.

#### 3.6.2. Determination of Fatty Acid Composition by GC–FID

Transmethylation of oil samples was performed in triplicate according to GB5009.168-2016. FAME standards were analyzed using a gas chromatograph. The GC system was an Agilent 7697A series (Agilent, Santa Clara, CA, USA) which was equipped with a HP-88 capillary column (60 m × 25 mm × 0.20 µm, Thermo Fisher, Grand Island, NY, USA) and flame ionization detector (FID, Agilent, Santa Clara, CA, USA). Nitrogen was used as a carrier gas with a constant flow rate of 1 mL/min. The injector and FID temperatures were kept at 250 °C. The oven temperature was set as 40 °C for 3 min initially, and was raised at 40 °C/min to 120 °C and held at that temperature for 2 min. Then, it was raised to 181 °C at a rate of 3 °C/min and held for 15 min. Finally, the oven temperature increased to 212 °C at the rate of 2 °C/min and held for 20 min. Standards of 37 mixed fatty acids were diluted into the corresponding individual stock solution by methylation and then mixed together (0.1 g/L). Retention times (RTs) of fatty acids were compared with those of authentic standards, and contents of individual components were expressed in relative percentages [32].

#### 3.6.3. Determination of Volatile Organic Compounds by GC–IMS

The GC–IMS instrument (Flavor spec^®^ H1-00053, Gesellschaft für Analytische Sensorsysteme mbH (G.A.S.), Dortmund, Germany) was equipped with a syringe and an auto-sampler unit for headspace analysis. The chromatographic separation was performed with a FS-SE-54-CB-1 capillary column (15 m, ID: 0.53 mm), a radioactive ionization source (tritium) of 6.5 KeV, and a heated splitless injector for direct automatic sampling of headspace volatile compounds from the oil samples into the GC–IMS.

The BDSO sample (2.5 mL) was heated at 80 °C for 10 min in an incubator box in order to generate volatile compounds from oil sample. The injection volume was set to 400 μL with an injection speed of 0.6 mL/s and a syringe temperature of 80 °C. The temperature of the automatic headspace was set to 85 °C for 15 min. After sample injection, the volatile organic compounds were pushed into the multi-capillary column through a carrier gas for time-based separation. The chromatographic separation was executed under isothermal conditions: the carrier gas flow was initially set at 2 mL/min during 2 min and the flow was linearly increased to 15 mL/min within 8 min; then it was raised to 80 mL/min within 10 min, and finally, the flow reached 150 mL/min in the next 5 min. The total run time was 40 min in order to achieve good separation effect.

After the separation in the capillary column at 60 °C, the headspace was pushed into the ionization chamber for prior ionization, then driven into the drift region via a shutter grid, and finally passed into the IMS detector. Operating conditions of IMS were set as follows: the drift tube length was 5 cm and operated at a constant voltage of 400 V/cm; the temperature of the drift tube was set to 45 °C with a nitrogen (99.999% purity) flow rate of 150 mL/min. The determination of headspace volatiles was performed in duplicate for each oil sample.

#### 3.6.4. Thermal Stability and Thermal Behavior Analysis

The thermal stability of the extracted seed oil was determined by thermogravimetric analysis (TG) by a simultaneous thermal analyzer (STA 449F3, Netzsch, Bavaria, Germany). The thermal behavior of each extracted oil sample was analyzed by a differential scanning calorimeter (DSC 200F3, Netzsch, Bavaria, Germany). Approximately 25.5 mg of the sample was heated from room temperature to 700 °C at a heating rate of 5 °C/min, and an empty pan was used as a standard. The sample and reference pans were placed inside the calorimeter, and inert nitrogen gas was employed for removal of oxygen at 20 mL/min at 10 °C/min.

### 3.7. Determination of In Vitro Antioxidant Activity

#### 3.7.1. DPPH Radical Scavenging Activity Assay

The DPPH radical scavenging activity was determined using the method described by Ruttarattanamongkol and Petrasch [55] with slight modification. Briefly, BDSO was diluted with ethanol/petroleum ether (1:1 v/v) to different concentrations (6.0, 5.0, 4.0, 3.0, 2.0, 1.0, and 0.5 mg/mL). Then, 1 mL of the diluted seed oil solution was added into 2 mL DPPH solution (0.2 mM in dehydrated alcohol). The mixture was shaken vigorously and incubated in the dark for 30 min at room temperature and then its absorbance was measured at 517 nm with a UV-Vis spectrophotometer (UV759, Shanghai Precision & Scientific Instrument Co., Ltd., Shanghai, China). The test was repeated three times, and the radical-scavenging activities of samples were calculated according to Equation (4).
(4)$$\mathrm{DPPH}\;\mathrm{radical}\;\mathrm{scavenging}\;\mathrm{activity}=\left(\frac{A_{control}-A_{sample}}{A_{control}}\right)\times 100\%$$
where *A_control_* is the absorbance at 517 nm of the DPPH solution without sample, and *A_sample_* is the absorbance of the DPPH solution with sample.

#### 3.7.2. ABTS Radical Scavenging Activity Assay

The ABTS radical scavenging activity assay was carried out by the method described by Aleksandar and Agnieszka [56] with some modifications. Briefly, 2.45 mmol/L potassium persulfate was mixed with 7 mmol/L aqueous ABTS to give the ABTS radical solution. The mixture was incubated in the dark for 16 h at room temperature. The ABTS radical solution was diluted with phosphate-buffered saline (PBS) (pH = 7.0) to an absorbance of 0.70 ± 0.02 at 734 nm. BDSO was diluted with ethanol/petroleum ether (1:1 *v*/*v*) to the different concentrations (6.0, 5.0, 4.0, 3.0, 2.0, 1.0, and 0.5 mg/mL).The sample solution was mixed with ABTS radical solution in the ratio of 1:20. The mixture was shaken vigorously and incubated in the dark for 6 min at room temperature, and then its absorbance was measured at 734 nm. The radical scavenging activities of samples were calculated according to Equation (5).
(5)$$\mathrm{ABTS}\;\mathrm{radical}\;\mathrm{scavenging}\;\mathrm{activity}=\left(\frac{A_{control}-A_{sample}}{A_{control}}\right)\times 100\%$$

*A_control_* is the absorbance at 734 nm of the ABTS solution without sample, and *A_sample_* is the absorbance of the ABTS solution with sample.

#### 3.7.3. Reducing Power Assay

The reducing power assay was performed according to Zeng et al. [57] with some modifications. BDSO was diluted to the different concentrations (6.0, 5.0, 4.0, 3.0, 2.0, 1.0, and 0.5 mg/mL). Then, 0.2 mmol/L phosphate buffer (pH = 6.6) was mixed with 1% KOH solution. Subsequently, 1.0 mL of the sample was added into the mixture and was incubated in a water bath at 50 °C for 20 min. Then, 10% trichloroacetic acid solution was added to the mixture to stop the reaction. The mixture was centrifuged at 5000 rpm for 10 min, and the supernatant was mixed with 0.1% ferric chloride solution. The absorbance was measured at 700 nm to obtain the reducing power activity.

### 3.8. Determination of In Vitro Cell-Viability Assay

#### 3.8.1. Preparation of BDSO Samples Obtained from SC-CO_2_ and PEE for Cellular Experiments

BDSO samples obtained from SC-CO_2_ and PEE were prepared for use in cellular experiments according to the previous report with some modification [58]. In brief, 18.75 µL of BDSO samples extracted by SC-CO_2_ and PEE were dissolved in 481.25 µL 100% ethanol, respectively, to reach a final concentration of 30 mg/mL. The solution was then mixed with 4.5 mL of 20% BSA in PBS at 50 °C for 1 h, yielding a final stock solution of 3 mg/mL. A solvent control (18% BSA) was prepared by mixing 481.25 µL of 100% ethanol with 4.5 mL of 20% BSA. All stock solutions were stored at −20 °C.

#### 3.8.2. Cell Culture

Six kinds of cell lines were used to evaluate the effects of seed oil obtained by SC-CO_2_ extraction and PEE on the viability of cells in vitro, including three lines of cancer cells (human HGC-27 gastric cancer cells, human SH-SY5Y neuroblastoma cells and rat C6 glioma cells) and three lines of normal cells (mouse 3T3-L1 preadipocytes, rat H9C2 cardiomyoblast cells, and human umbilical vein endothelial cells). Cells were routinely cultured in DMEM medium with 10% fetal bovine serum (FBS) and 1% penicillin–streptomycin in an incubator under 5% CO_2_ atmosphere at 37 °C. HUVEC cells were cultured in ECM medium supplemented with 1% ECGS.

#### 3.8.3. Cellular Viability Assessment

Cellular viability was determined by the MTT assay [59]. Cells were seeded overnight in 96-well plates at a density of 1 × 10^5^ cells/well overnight. Then the cells were treated with different concentrations (1–200 µg/mL) of BDSO obtained by SC-CO_2_ and PEE, respectively, for another 24 h. Untreated cells (normal group) and solvent control cells (control group) were included as controls. At the end of the incubation with samples for 24 h, MTT solution (5 mg/mL) was added to each well and the cells were incubated in the dark for 4 h. Then MTT solution was removed and the formazan product was solubilized in 150 µL of DMSO, and further incubated for 15 min at 37 °C. The absorbance was measured at 490 nm by a fluorescence microplate reader to determine cell viability. Each experiment was performed in triplicate.

### 3.9. Statistical Analysis

Gas chromatography–ion mobility spectrometry analysis results were plotted in a 3D graph. The visual spectra comparisons of volatile flavor compounds between samples were obtained by IMS Control LAV software version 2.2.1 (Gesellschaft für Analytische Sensorsysteme mbH (G.A.S.), Dortmund, Germany, 2017). The identification of specific volatile compounds was realized by the software GC × IMS Library Search version 1.0.3 which were obtained from G.A.S. (Gesellschaft für Analytische Sensorsysteme mbH (G.A.S.), Dortmund, Germany, 2017). All the tests were repeated three times and the data were expressed as mean ± SD. The data were analyzed using SPSS 19.0 (SPSS Inc., IBM Company, Chicago, IL, USA, 2010).

## 4. Conclusions

Supercritical carbon dioxide (SC-CO_2_) extraction and solvent extraction methods were employed to extract oil from *B. dasystachya* seeds. The extraction conditions and the yield were optimized, and the properties of the oil extracts were characterized. The results showed that the maximum oil yield of 12.54 ± 0.56 g/100 g (w/w, *n* = 4) could be achieved at a pressure of 25.00 MPa, a temperature of 59.03 °C, and a CO_2_ flow rate of 2.25 SL/min. The GC results illustrated that the oil extracted using the SC-CO_2_ method contained approximately 57.90% PUFAs, with high amounts of linoleic acid (18:2) and linolenic acid (18:3). The PUFA content in oil extracted by the SC-CO_2_ method was also higher than that extracted by the PEE method. This could be due to the fact that supercritical carbon dioxide extraction is more effective as this process reduces the thermal degradation and oxidation of unsaturated fatty acids. SC-CO_2_ extracts exhibited higher antioxidant ability in the test system, with higher levels of PUFAs. The result from the in vitro cytotoxicity assay further indicated that the oil extracted by both methods was not toxic to either normal or cancer cells. Considering its high extraction yield and the fine quality of the oil produced, the SC-CO_2_ extraction method is an effective and efficient technique for extracting oil from *B. dasystachya* seeds. The optimization of extraction conditions and the determination of fatty acid composition presented in this work can be applied in or used as a starting point for future industrial applications.

## Figures and Tables

**Figure 1 molecules-25-01836-f001:**
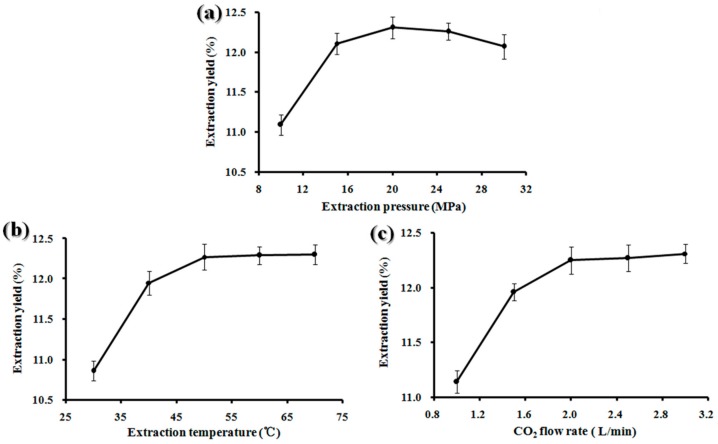
*Berberis dasystachya* seed oil yield with different extraction parameters. (**a**) Extraction pressure; (**b**) Extraction temperature; (**c**) CO_2_ flow rate.

**Figure 2 molecules-25-01836-f002:**
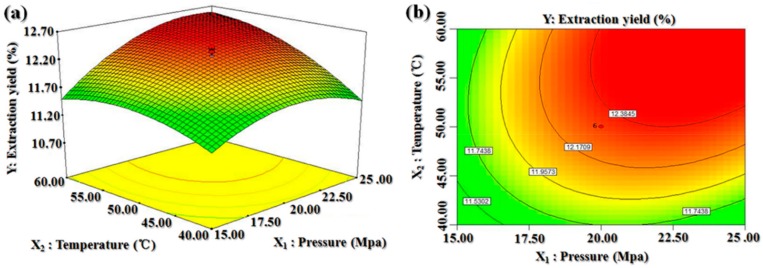
Response surface (**a**) and contour plots (**b**) for the effect of pressure and temperature on the *B. dasystachya* seed oil yield.

**Figure 3 molecules-25-01836-f003:**
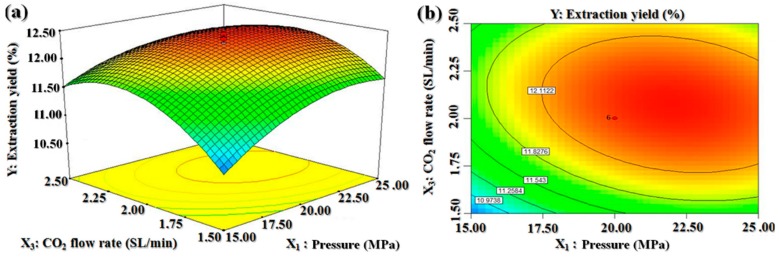
Response surface (**a**) and contour plots (**b**) for the effect of pressure and CO_2_ flow rate on the *B. dasystachya* seed oil yield.

**Figure 4 molecules-25-01836-f004:**
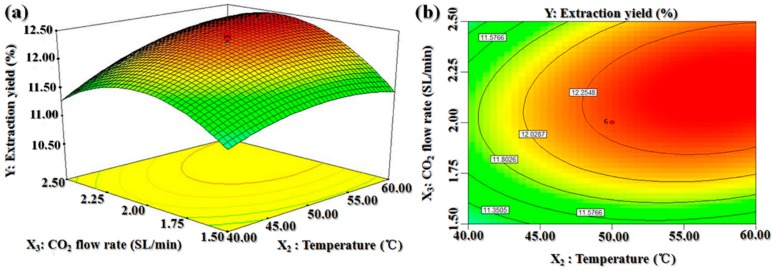
Response surface (**a**) and contour plots (**b**) for the effect of temperature and CO_2_ flow rate on the *B. dasystachya* seed oil yield.

**Figure 5 molecules-25-01836-f005:**
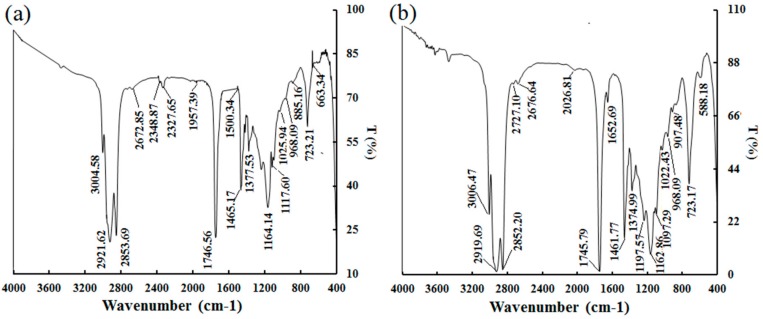
FT-IR spectra of *B. dasystachya* seed oil obtained using different methods: (**a**) The supercritical carbon dioxide (SC-CO_2_) extraction method; (**b**) The petroleum ether extraction (PEE) method.

**Figure 6 molecules-25-01836-f006:**
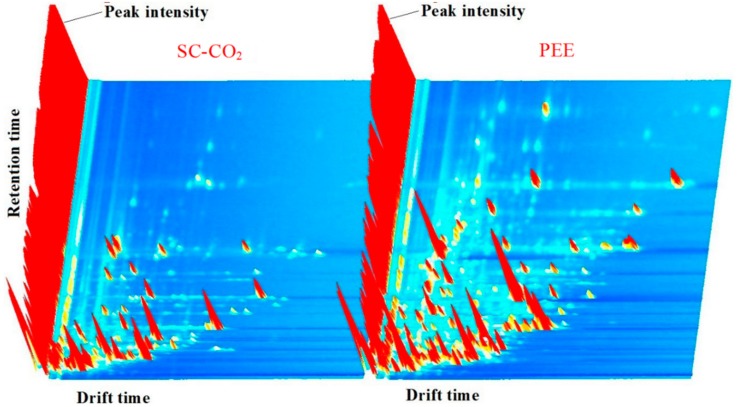
Three-dimensional topographic images of *B. dasystachya* seed oil obtained by the SC-CO_2_ and PEE methods.

**Figure 7 molecules-25-01836-f007:**
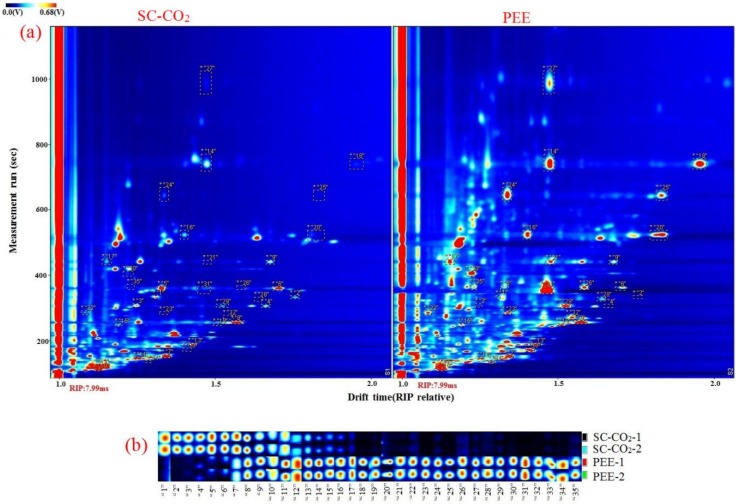
Selection locations of 35 characteristic peaks (spots) (**a**), and an overview (**b**) of the peaks of *B. dasystachya* seed oil obtained by the SC-CO_2_ and PEE methods.

**Figure 8 molecules-25-01836-f008:**
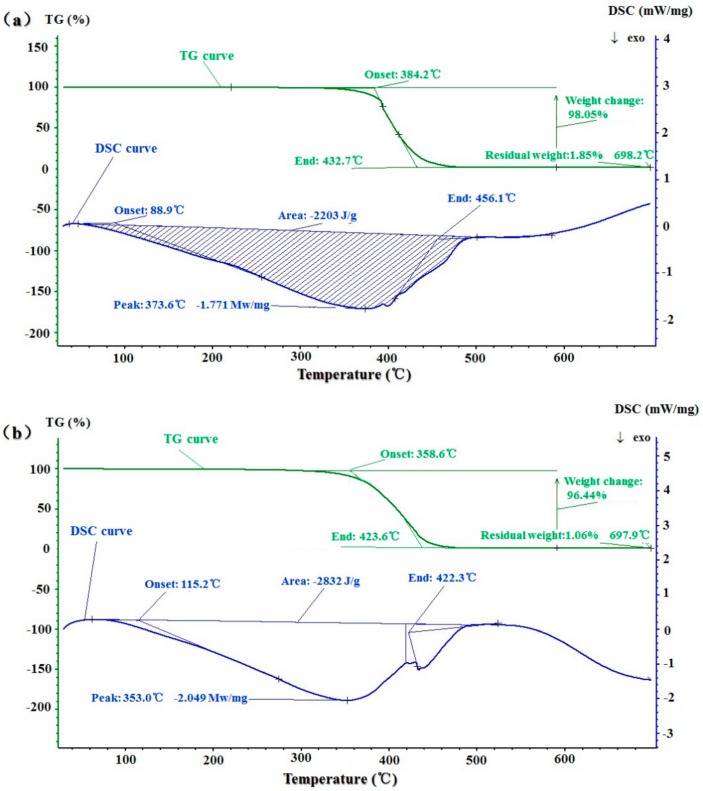
Thermogravimetric and differential scanning calorimetry (DSC) curves of *B. dasystachya* seed oil obtained by different methods: (**a**) The SC-CO_2_ extraction method; (**b**) The PEE method.

**Figure 9 molecules-25-01836-f009:**
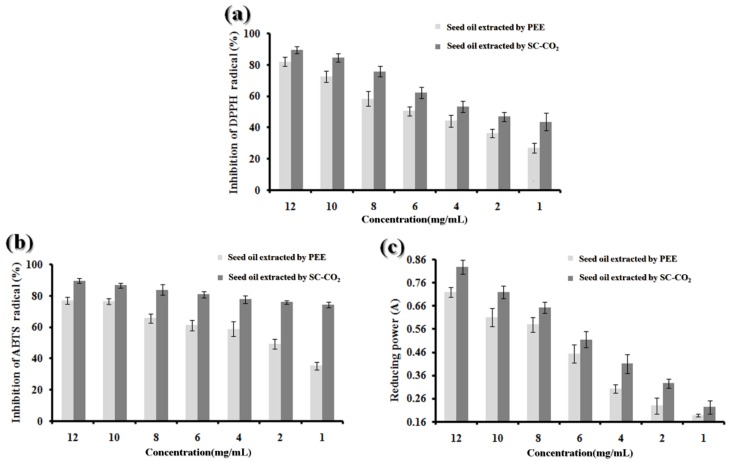
Activity of *B. dasystachya* seed oil obtained by different methods at different concentrations. (**a**) DPPH scavenging activity; (**b**) ABTS scavenging activity; (**c**) reducing power. Data are means ± SD (*n* = 3).

**Figure 10 molecules-25-01836-f010:**
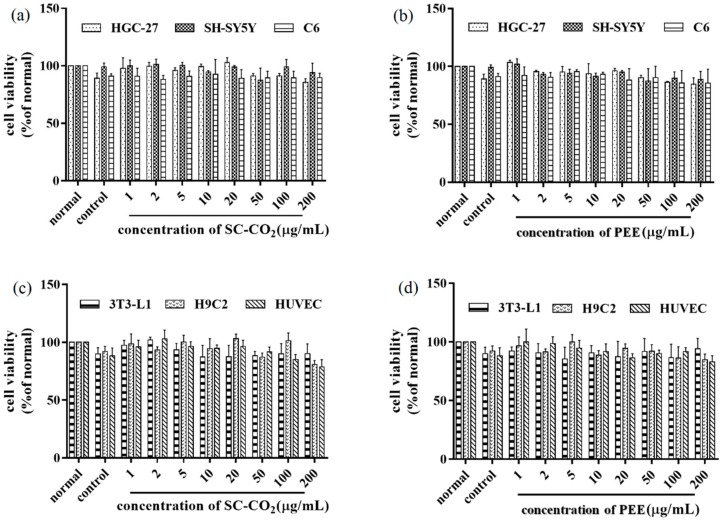
The effects of *B. dasystachya* seed oil obtained by different methods on the viability of cells in vitro. (**a**) Oil extracted using SC-CO_2_ on tumor cells; (**b**) Oil extracted using PEE on tumor cells; (**c**) Oil extracted using SC-CO_2_ on normal cells; (**d**) Oil extracted using PEE on normal cells. Data are given as means ± SD (*n* = 3).

**Table 1 molecules-25-01836-t001:** Central composite design and results for the yield of *B. dasystachya* seed oil (BDSO).

Run	Pressure (*X_1_*) (MPa)	Temperature (*X_2_*) (°C)	CO_2_ Flow Rate (*X_3_*) (SL/min)	BDSO Yield (%)	
				Measured Value	Predicted Value
1	−1	−1	−1	10.39	11.23
2	1	1	1	12.44	12.35
3	1.682	0	0	11.89	11.77
4	0	0	0	12.42	12.35
5	1	1	−1	11.56	11.69
6	1	−1	1	10.72	11.08
7	−1	1	−1	10.66	10.36
8	0	0	0	12.36	12.35
9	−1	−1	1	11.07	10.34
10	0	0	1.682	10.95	11.77
11	0	0	0	12.44	12.35
12	0	1.682	0	12.02	12.07
13	−1.682	0	0	10.74	10.78
14	0	0	0	12.35	12.35
15	0	0	0	12.25	12.35
16	0	0	0	12.29	12.35
17	0	0	−1.682	10.41	10.38
18	0	−1.682	0	11.18	10.96
19	1	−1	−1	10.98	10.90
20	−1	1	1	11.39	11.76

**Table 2 molecules-25-01836-t002:** Regression coefficients of the predicated second-order polynomial model for the response variable.

Standard Deviation	Coefficient of Variation (CV, %)	Adjusted R^2^	Adeq Precision	Predicted R^2^	R^2^
0.16	1.38	0.9560	17.878	0.8679	0.9768

**Table 3 molecules-25-01836-t003:** ANOVA for the response surface quadratic model.

Source	Sum of Squares (SS)	Degree of Freedom (df)	Mean Square (MS)	F-Value	*p*-Value (Prob > F)
*Model*	10.72	9	1.19	46.84	<0.0001 **
*X_1_*	1.31	1	1.31	51.39	0.0001 **
*X_2_*	1.42	1	1.42	55.83	<0.0001 **
*X_3_*	0.77	1	0.77	30.20	0.0003 **
*X_1_X_2_*	0.41	1	0.41	16.11	0.0025 **
*X_1_X_3_*	0.21	1	0.21	8.18	0.0169 *
*X_2_X_3_*	0.28	1	0.28	10.92	0.00080 **
*X_1_X_1_*	1.74	1	1.74	68.31	<0.0001 **
*X_2_X_2_*	0.87	1	0.87	34.41	0.0002 **
*X_3_X_3_*	4.71	1	4.71	185.24	<0.0001 **
Residual	0.25	10	0.025	-	-
Lack of Fit	0.16	5	0.033	1.85	0.2587
Pure Error	0.089	5	0.018	-	-
Cor Total	10.97	19	-	-	-

** Significant at *p* < 0.01; * Significant at *p* < 0.05.

**Table 4 molecules-25-01836-t004:** Predicted and experimental values of the responses at optimum conditions.

Pressure (*X_1_*) (MPa)	Temperature (*X_2_*) (°C)	CO_2_ Flow Rate (*X_3_*) (SL/min)	*B. dasystachya* Seed Oil Yields (g/100 g)	
			Actual Value	Predicted Value
25	59.03	2.25	12.54 ± 0.56	12.553

**Table 5 molecules-25-01836-t005:** Fatty acid composition and content of *B. dasystachya* seed oil by PEE and SC-CO_2_ extraction (area%, *n* = 3).

**SFA**	**C11**	**C13**	**C14**	**C15**	**C16**	**C18**	**C20**	**∑SPA**
PEE ^a^	0.11 ± 0.02	0.24 ± 0.04	3.65 ± 0.63	-	12.19 ± 1.12	2.03 ± 0.44	0.90 ± 0.16	19.12
SC-CO_2_ ^b^	0.15 ± 0.01	0.18 ± 0.03	3.69 ± 0.77	0.47 ± 0.06	5.98 ± 1.33	3.16 ± 0.57	0.74 ± 0.09	14.37
**UFA**	**C16:1**	**C18:1**	**C18:2**	**C18:3**	**C22:1**	**∑UFA**	**∑PUFA**	**SFA/UFA**
PEE ^a^	0.63 ± 0.14	23.08 ± 1.58	21.41 ± 1.09	32.28 ± 2.02	3.47 ± 0.25	80.87	53.69	0.2364
SC-CO_2_ ^b^	1.77 ± 0.32	20.34 ± 1.30	23.16 ± 1.97	34.74 ± 1.91	5.61 ± 0.83	85.62	57.90	0.1678

^a^ BDSO obtained by petroleum ether extraction at 80 ℃ for 7 h; ^b^ BDSO obtained by supercritical CO_2_ fluid extraction at 25.00 MPa, 59.0 ℃, 2.25 SL/min. C11: undecanoic acid, C13: tridecanoic acid, C14: myristic acid, C15: pentadecanoic acid, C16: palmitic acid, C18: stearic acid, C20: eicosanoic acid; C16:1: palmitoleic acid, C18:1: oleic acid, C18:2: linoleic acid, C18:3: linolenic acid, C22:1: erucic acid. SFA: saturated fatty acid, UFA: unsaturated fatty acid, PUFA: polyunsaturated fatty acid.

**Table 6 molecules-25-01836-t006:** Volatile organic compounds and content of *B. dasystachya* seed oil extracted by PEE and SC-CO_2_.

Marker Number	Volatile Organic Compounds	Chemical Abstracts Service (CAS#)	Retention Index	Retention Time (s)	Drift Time (ms)	Formula	Comment	Peak Intensity	Peak Intensity
								PEE ^a^	SC-CO_2_ ^b^
**Esters**									
1	Isoamyl acetate	C123922	877.3	333.077	1.3124	C_7_H_14_O_2_	Monomer	-	271.3 ± 15.03
2	Isoamyl acetate	C123922	876.7	332.413	1.7576	C_7_H_14_O_2_	Dimer	-	166.7 ± 7.02
3	Ethyl isovalerate	C108645	854.2	308.514	1.2494	C_7_H_14_O_2_	Monomer	31.33 ± 1.54	394.0 ± 10.53
4	Ethyl isovalerate	C108645	852.2	306.523	1.664	C_7_H_14_O_2_	Dimer	66.32 ± 1.87	248.43 ± 8.02
7	Ethyl acetate	C141786	611.5	153.041	1.3413	C_4_H_8_O_2_	Monomer	989.0 ± 4.58	2176.01 ± 46.36
27	Butyl hexanoate	C626824	1176.6	985.551	1.4731	C_10_H_20_O_2_	Monomer	412.3 ± 2.08	37.50 ± 1.43
32	Ethyl lactate	C97643	815	271.866	1.5439	C_5_H_10_O_3_	Monomer	551.50 ± 4.58	-
**Aldehydes**									
5	Heptanal	C111717	901.2	361.114	1.3312	C_7_H_14_O	Monomer	304.33 ± 1.53	862.23 ± 28.09
6	Heptanal	C111717	900.1	359.762	1.7035	C_7_H_14_O	Dimer	443.56 ± 1.57	1070.68 ± 18.24
8	Hexanal	C66251	796.4	256.461	1.5682	C_6_H_12_O	Monomer	1877.63 ± 11.06	2000.56 ± 15.54
9	*t*-2-heptenal	C18829555	958.8	440.986	1.6788	C_7_H_12_O	Monomer	304.76 ± 1.15	252.51 ± 13.65
13	Pentanal	C110623	694.3	189.878	1.431	C_5_H_10_O	Monomer	1172.30 ± 1.52	665.25 ± 22.34
14	*n*-Nonanal	C124196	1100.6	740.792	1.4731	C_9_H_18_O	Monomer	756.21 ± 5.50	302.11 ± 4.36
16	Octanal	C124130	1007.7	525.295	1.4019	C_8_H_16_O	Monomer	598.06 ± 15.71	174.48 ± 2.09
17	Benzaldehyde	C100527	958.3	440.108	1.1555	C_7_H_6_O	Monomer	519.57 ± 17.06	103.02 ± 5.67
18	*n*-Nonanal	C124196	1101.3	742.815	1.9518	C_9_H_18_O	Dimer	839.77 ± 7.02	27.09 ± 0.58
20	Octanal	C124130	1006.3	522.661	1.8334	C_8_H_16_O	Dimer	729.35 ± 5.86	-
21	Benzaldehyde	C100527	958.8	440.986	1.4756	C_7_H_6_O	Dimer	574.41 ± 8.32	-
22	Furfural	C98011	831.5	286.572	1.0841	C_5_H_4_O_2_	Monomer	623.59 ± 7.04	69.28 ± 6.65
23	Furfural	C98011	830.3	285.522	1.3376	C_5_H_4_O_2_	Dimer	871.76 ± 13.22	-
24	*t*-2-octenal	C2548870	1063.9	645.72	1.337	C_8_H_14_O	Monomer	752.24 ± 9.61	71.03 ± 5.29
25	*t*-2-octenal	C2548870	1063	643.698	1.8288	C_8_H_14_O	Dimer	524.25 ± 1.27	-
33	3-methylbutanal	C590863	660.8	173.425	1.4074	C_5_H_10_O	Monomer	956.55 ± 3.21	78.45 ± 4.06
34	2-methyl-propanal	C78842	559.5	135.569	1.2894	C_4_H_8_O	Monomer	783.14 ± 3.98	-
**Ketones**									
12	Acetone	C67641	505.7	121.009	1.1222	C_3_H_6_O	Monomer	2407.02 ± 35.78	1640.79 ± 24.31
15	2-Hexanone	C591786	787	249.108	1.1906	C_6_H_12_O	Monomer	397.71 ± 5.86	157.04 ± 8.08
19	2-Hexanone	C591786	787	249.108	1.5089	C_6_H_12_O	Dimer	781.09 ± 4.53	49.17 ± 1.92
31	Cyclohexanone	C108941	894.7	353.164	1.4612	C_6_H_10_O	Monomer	3594.16 ± 4.56	-
**Alcohols**									
10	Ethanol	C64175	467.6	112.478	1.1279	C_2_H_6_O	Monomer	2822.06 ± 35.77	2406.98 ± 42.50
11	1-Propanol	C71238	592	146.039	1.2509	C_3_H_8_O	Monomer	1483.37 ± 14.03	946.48 ± 15.14
26	2-Butoxyethanol	C111762	902.6	362.824	1.5871	C_6_H_14_O_2_	Monomer	638.25 ± 1.19	-
28	1-Hexanol	C111273	872.1	327.373	1.6389	C_6_H_14_O	Monomer	174.51 ± 2.64	-
29	2-Hexenol	C2305217	852.5	306.88	1.5264	C_6_H_12_O	Monomer	1471.66 ± 5.89	135.05 ± 3.82
**Terpenes**									
30	α-Pinene	C80568	935.2	405.857	1.2239	C_10_H_16_	Monomer	1102.94 ± 13.57	27.61 ± 2.07
**Organic acids**									
35	Pentanoic acid	C109524	903.3	363.627	1.2299	C_5_H_10_O_2_	Monomer	403.03 ± 4.09	29.58 ± 2.16

^a^ BDSO obtained by petroleum ether extraction at 80 °C for 7 h; ^b^ BDSO obtained by supercritical CO_2_ fluid extraction at 25.00 MPa, 59.0 °C, 2.25 SL/min.

**Table 7 molecules-25-01836-t007:** Independent variables and their levels used in the response surface method.

Independent Variables	Level				
	−1.682	−1	0	1	1.682
Pressure (*X_1_*, MPa)	11.59	15	20	25	28.41
Temperature (*X_2_*, ℃)	33.18	40	50	60	66.32
CO_2_ flow rate (*X_3_*, SL/min)	1.16	1.5	2.0	2.5	2.84

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
