# Peer review of "Characterization and Biological Activities of Seed Oil Extracted from Berberis dasystachya Maxim. by the Supercritical Carbon Dioxide Extraction Method"

_molecules, 2020, doi:10.3390/molecules25081836_

Round 1

Reviewer 1 Report

Manuscript is relatively well written and useful in the field of food technology and bioactive compounds in plant materials. I propose to change description of "SE" by petroleum ether extraction (PEE). Please correct unit Mpa - in whole text should be MPa. In line 497 please give real values of independent values (eg. pressure 11.59-28.41 MPa). Please explain/correct scale of axes in figures 2 -4 (now, it seems to be the range form -1 to +1). Please re-analyse your statement in lines 235-237; in my opinion results of this study do not confirm this hypothesis. Please re-analyse your discussion in lines 261-272. Especially, please to compare oil yield by petroleum ether and SC-CO2 extraction (in my opinion it can be the reason of obtained variation in oil recovery and variation in fatty acids shares).

In conclusion, Authors should briefly summarize posible reasons of stated variation of both oils BDSO; for example which oil components can significantly affect antioxidative activities of both oils.

Author Response

Dear Reviewer:

Thank you very much for your letter of 01-April-2020 informing us the conditional acceptance of our manuscript, “Characterization and biological activities of seed oil extracted from Berberis dasystachya Maxim. by supercritical carbon dioxide extraction method” (Manuscript ID:molecules-763959), together with the highly valuable comments, suggestions, and instructions from the editor and reviewers. We went through the letter carefully and greatly appreciated the editor and reviewers for the constructive comments, helpful suggestions, and detailed instructions.

A point-to-point response to the comments, suggestions, and instructions from the editor and reviewers were involved in this letter as following.  Explanations, corrections/revisions, and investigations based on the comments and suggestions from Reviewer 1: Comment 1: I propose to change description of "SE" by petroleum ether extraction (PEE).Response: Thanks a lot for the comments and suggestions. According to your reminder, we have corrected the mistakes in manuscript. We have revised the whole manuscript carefully and tried to avoid any error of the manuscript, the “SE” was revised as “PEE”, “Solvent extraction” was revised as “Petroleum ether extraction”. We have highlighted with red colored in our manuscript.

Comment 2: Please correct unit Mpa - in whole text should be MPa.

Response: Thanks a lot for the comments and suggestions. According to your reminder, we have corrected the mistake in our manuscript, in Figure 2, Figure 3,line 130, line 531, line 540, line 548, the unit “Mpa” was revised as “MPa”. We also have highlighted with red colored in our manuscript.

Comment 3: In line 497 please give real values of independent values (eg. pressure 11.59-28.41 MPa).

Response: Thanks a lot for the comments and suggestions. We are very sorry for our incorrect writing. According to your reminder, we have corrected the mistake in our manuscript and have highlighted with red colored in our manuscript. In line 546, page 21, the sentence “During extraction, the extraction time was set as 50 min, the pressure, temperature, and the CO2 flow rate were ranged between 15 and 25 MPa, 40 and 60℃, 1.5 and 2.5 SL/min, respectively.” was revised as “During extraction, the extraction time was set as 50 min. The pressure, temperature, and CO2 flow rate ranged between 11.59 and 28.41 MPa, 33.18 and 66.32℃, 1.16 and 2.84 SL/min, respectively.”. Comment 4: Please explain/correct scale of axes in figures 2 -4 (now, it seems to be the range form -1 to +1).Response: Thanks a lot for the comments and suggestions. The figures 2 -4 were derived from Design-Expert 7.1.3 Trial, State-Ease, Inc., Minneapolis, USA. We have check the scale of axes in figures 2-4, we are very sorry for our incorrect writing and  we have corrected the range of ordinate axis in figures 3-4.

Figure 3. Response surface (a) and contour plots (b) for the effect of pressure and CO2 flow rate on the B. dasystachya seed oil yield   ï¼ˆPlease see the manuscript.)

Figure 4. Response surface (a) and contour plots (b) for the effect of temperature and CO2 flow rate on the B. dasystachya seed oil yield(Please see the manuscript.)

Comment 5: Please re-analyse your statement in lines 235-237; in my opinion results of this study do not confirm this hypothesis.Response: Thanks a lot for the comments and suggestions. We have re-examined this statement, and the results of seed oil yield cannot explain the statement: “we can conclude that using SC-CO2 extraction can extract BDSO with superior quality and higher concentration of active compounds compared with that extracted by the conventional PEE methods as has also been described previously.”

We are very sorry for our incorrect writing. According to your reminder, we have corrected the mistake in our manuscript. In line 242, page 9, the statement was revised as: “In addition, it is worth noting that BDSO extracted by the PEE method contained traces of organic solvents. The solvent residue left in the oil is considered to be toxic to humans and animals. Therefore, other operational steps such as evaporation concentration, nitrogen concentration, etc. are needed after the extraction completed to obtain the required samples. These steps will lead to the loss of oil yield. SC-CO2 extraction is efficient, safe, and can reduce the use of organic solvents for oil extraction. SC-CO2 method could extract BDSO with superior extraction yield compared to the conventional PEE methods, as described previously [17, 31]. Thus, we can conclude that SC-CO2 extraction method is an acceptable and environmentally friendly procedure for oil extraction from B. dasystachya seeds. ”.

  • Belbaki, A., Louaer, W., Meniai, A.H. Supercritical CO2 extraction of oil from Crushed Algerian olives. J Supercrit Fluid. 2017, 130, 165-171.

 Comment 6: Please re-analyse your discussion in lines 261-272. Especially, please to compare oil yield by petroleum ether and SC-CO2 extraction (in my opinion it can be the reason of obtained variation in oil recovery and variation in fatty acids shares). Response: Thanks a lot for the comments and suggestions. According to your reminder, we have reanalyze the discussion about variation in fatty acids shares in our manuscript (in line 289, page 10) and have highlighted with red colored in our manuscript. The composition and the ratio of saturated/unsaturated fatty acid (SFA/UFA) are indicators that can be used to evaluate nutritional and functional characteristics of oils. The SFA/UFA ratio of BDSO obtained from the PEE and the SC-CO2 methods were 0.2364 and 0.1678, respectively. BDSO obtained by the SC-CO2 method was rich in both UFA (nearly 85.62% of the total fatty acids) and polyunsaturated fatty acids (PUFA; nearly 57.90% of the total fatty acids). Both the UFA and PUFA contents obtained in the oil extracted by the PEE method were lower than those in oil extracted by the SC-CO2 method. According to the reference, there may be three reasons for this result. Firstly, SC-CO2 extraction method has superior BDSO yield compared with that extracted by the conventional PEE methods. Secondly, in the extraction of fatty acids using an organic solvent, the samples are concentrated using a rotary evaporator at high temperature before direct injection, and are further reduced under a stream of N2 [34]. These concentration steps require the use of high temperatures, which leads to the thermal degradation of bioactive compounds such as polyunsaturated fatty acids [35]. By contrast, in the extraction of fatty acids by the SC-CO2 method, the oil is transferred directly into the injection port of the GC or GC-MS system for analysis [36], and the extraction process is carried out at low temperatures, which can minimize thermal damage of UFA and PUFA in the oil. Furthermore, carbon dioxide reaches the critical point under relatively mild conditions, and extraction in the absence of oxygen is beneficial for the preservation of bioactive compounds (PUFA, UFA) without oxidation [37]. Based on the above, we can conclude that SC-CO2 extraction can extract BDSO with superior quality and higher concentration of polyunsaturated fatty acids compared to the conventional PEE methods.  

  • El Asbahani, A., Miladi, K., Badri, W., Sala, M., Addi, E.H.A. Casabianca, H., El Mousadik, A., Hartmann, D., Jilale, A., Renaud, F.N.R., Elaissari, A. Essential oils: from extraction to encapsulation. J. Pharm. 2015, 483: 220-243.
  • Jiang, M. H., Yang, L., Zhu, L., Piao, J.H., Jiang, J.G. Comparative GC/MS analysis of essential oils extracted by 3 methods from the bud of Citrus aurantium var. amara Engl. J. Food Sci. 2011, 76, 1219-1225..
  • Temelli, F. Perspectives on supercritical fluid processing of fats and oils. J Supercrit Fluid. 2009, 47: 583-590.

Comment7: In conclusion, Authors should briefly summarize possible reasons of stated variation of both oils BDSO; for example which oil components can significantly affect antioxidative activities of both oils.

Response: Thanks a lot for the comments and recommendations. We have revised the conclusion and highlighted with red colored in our manuscript (in line 675, page 24).

Supercritical carbon dioxide (SC-CO2) extraction and solvent extraction methods were employed to extract oil from B. dasystachya seeds. The extraction conditions and the yield were optimized, and the properties of the oil extracts were characterized. The results showed that the maximum oil yield of 12.54 ± 0.56 g/100 g (w/w, n=4) could be achieved at a pressure of 25.00 MPa, a temperature of 59.03℃, and a CO2 flow rate of 2.25 SL/min. The GC results illustrated that the oil extracted using the SC-CO2 method contained approximately 57.90% PUFA with high amounts of linoleic acid (18:2) and linolenic acid (18:3). The PUFA content in oil extracted by the SC-CO2 method was also higher than that extracted by the PEE method. This could be due to the fact that supercritical carbon dioxide extraction is more effective as this process reduces the thermal degradation and oxidation of unsaturated fatty acids. SC-CO2 extracts exhibited the higher antioxidant ability in the test system with higher levels of PUFA. The result from in vitro cytotoxicity assay further indicated that the oil extracted by both methods was not toxic to both normal and cancer cells. Considering its high extraction yield and the fine quality of oil produced, the SC-CO2 extraction method is an effective and efficient technique for extracting oil from B. dasystachya seeds. The optimization of extraction conditions and the determination of fatty acids composition presented in this work can be applied in or used as a starting point for future industrial applications.

Comment 8: English language and style are fine/minor spell check required.

Response: Thanks a lot for the comments and recommendation. We are very sorry for our incorrect writing. According to your reminder, we have corrected the mistakes in manuscript. We have revised the whole manuscript carefully and tried to avoid any error of the manuscript, we have highlighted with green colored in our manuscript. We very much appreciate your valuable suggestions and kind advice. We also noticed that we have made some mistake in the subtitles, we also revised it. We have rewritten the part of Statistical Analysis and Funding and highlighted with green colored words in manuscript as well. We highly value this opportunity and we have endeavored to revise the manuscript according to the instructions, comments, and suggestions from the editor and reviewers, necessary explanations, and corrections/revisions have been done carefully to improve the quality of the manuscript. The changes in the revised manuscript and supporting information have been highlighted in different color.

We are very grateful to you for your kind advice, helpful instructions and continuous effort in the processing of our manuscript and we will be very happy to provide any further information if needed.

Yours sincerely,Lijuan Han and Gang Li, DrQinghai UniversityXining, China, 810001Tel: +86-13678659123E-mail: [email protected];  [email protected]

Reviewer 2 Report

Molecules (Manuscript ID: molecules-763959), Comments to the Authors:

Title: Characterization and biological activities of seed oil extracted from Berberis dasystachya Maxim. by supercritical carbon dioxide extraction method

Comments

The submitted manuscript discussed the use of the central composite design (CCD) combined with response surface methodology to optimize the extraction conditions of B. dasystachya oil (BDSO) using supercritical carbon dioxide (SC-CO2) extraction method, and the results were compared with those obtained by the traditional solvent extraction (SE) method. A variety of chemical characteristics of BDSO were analyzed, antioxidant and cellular viability in vitro have been studied by DPPH, ABTS, reducing power assay, and MTT assay, respectively. The results showed that the maximum yield of 12.54 ± 0.56 g / 100 g was obtained at the optimal extraction conditions, which were: pressure, 25.00 MPa; temperature 59.03 °C; and CO2 flow rate, 2.25 SL/min. The GC analysis results showed that BDSO extracted by the SC-CO2 method had higher contents of unsaturated fatty acids (85.62%) and polyunsaturated fatty acids (57.90%) than that extracted by the SE method. The GC-IMS results showed that the main volatile compounds in BDSO were aldehydes and esters. BDSO also exhibited antioxidant ability in a dose-dependent manner, and normal and cancer cells incubated with BDSO had survival rates of more than 85%, which indicates that BDSO is not cytotoxic.

I think the manuscript can be accepted for publication after the authors respond to the following comments:

  1. Table 5 has no meaning in the text. It should be moved to supporting information
  2. The authors should state the name of the fatty acids in table 6 and move it to the supporting information.
  3. The authors should compare their results with previous reports such as Antioxidant and Anti-fatigue Activities of Seed Oil from the Berries of Three Indigenous Plants in Tibetan Plateau Lijuan Han, JingMeng, Yongjing Yang, Ying Ye, Yourui Suo, Journal of Food and Nutrition Research, 2015, Vol. 3, No. 7, 445-457.

Author Response

Dear Reviewer:

Thank you very much for your letter of 01-April-2020 informing us the conditional acceptance of our manuscript, “Characterization and biological activities of seed oil extracted from Berberis dasystachya Maxim. by supercritical carbon dioxide extraction method” (Manuscript ID:molecules-763959), together with the highly valuable comments, suggestions, and instructions from the editor and reviewers. We went through the letter carefully and greatly appreciated the editor and reviewers for the constructive comments, helpful suggestions, and detailed instructions.

A point-to-point response to the comments, suggestions, and instructions from the editor and reviewers were involved in this letter as following.  Explanations, corrections/revisions, and investigations based on the comments and suggestions from Reviewer 2: Comment 1: Table 5 has no meaning in the text. It should be moved to supporting Information.Response: According to the reviewer’s reminder, we have deleted Table 5 and added the corresponding information in manuscript. We have highlighted these corrections with blue colored in our manuscript (in line 252, page 9). 

Figure 5 shows the Fourier transform infrared (FT-IR) spectrum of BDSO. Various functional groups present in the BDSO obtained by different methods were identified from the FT-IR analysis as shown in Table 5. The absorption bands of BDSO at 2348.87 and 2327.65 cm-1(Figure 5 a) were assigned to bending vibration of C≡ stretch, which clearly indicates that BDSO contains CO2. This is in agreement with previously reported results [32]. The absorption band of BDSO at 1500.34 cm-1(Figure 5 a)were assigned to C=C stretch on benzene ring, which indicates that BDSO obtained by SC-CO2 contains benzenoid hydrocarbon. The absorption bands of BDSO at 1465.17 cm-1(Figure 5 a), 1461.77 cm-1 (Figure 5 b)and 1745 cm-1 were assigned to bending vibration of lipid CH2 groups and the ester carbonyl stretching (C=O) of fatty acids, respectively[32]. The FT-IR spectrum clearly indicates that BDSO contains =C–H, C=C, and -C=C=C- groups, suggesting that the oil is unsaturated fatty acid containing ester groups [33]. The absorption bands at 1745 and 900~1200 cm−1 indicate the presence of ester groups, suggesting that the oil obtained from different extraction methods can be transformed into other types of ester, which can serve as a low-grade feedstock for biodiesel synthesis. However, compared with the previous research results [12], the infrared spectra of the three kinds of seed oil (BDSO obtained by screw pressing, PEE, SC-CO2, respectively) have similar characteristic absorbance bands such as hydroxyl groups(around 3000 cm-1),  -CH2 (around 1460 cm-1), -CH3 (around 1377 cm-1), and - (CH2)n- (n≥4, around 720 cm-1).

Comment 2: The authors should state the name of the fatty acids in table 6 and move it to the supporting information.Response: Thanks a lot for the comments and recommendations. According to the reviewer’s reminder, we have state the name of the fatty acids in Table 5(in line 312, page 11) and added the supporting information in manuscript (in line 276, page 10). We have highlighted these corrections with blue colored in our manuscript.

Table 5 presents the fatty acid compositions of BDSO extracted by PEE and SC-CO2 extraction methods. The fatty acids in BDSO extracted by the two methods mainly included saturated fatty acids (SFA) such as undecanoic acid (C11), tridecanoic acid (C13), myristic acid (C14), pentadecanoic acid (C15), palmitic acid (C16), stearic acid (C18), and eicosanoic acid (C20), and unsaturated fatty acids (UFA) such as palmitoleic acid (C16:1), oleic acid (C18:1), linoleic acid (C18:2), linolenic acid (C18:3), and erucic acid (C22:1). In the previous study [12], the fatty acid content of BDSO obtained by screw pressing was analyzed by GC-MS, and the result was different from that of GC analyses in this paper: six saturated fatty acids (lauric acid, myristic acid, pentadecanoic acid, heptadecylic acid, stearic acid) and six unsaturated fatty acids (palmitoleic acid, oleic acid, linoleic acid, γ-linolenic acid, α-linolenic acid, eicosenoic acid) were identified. Furthermore, it was found that linolenic acid had the highest unsaturated fatty acid content in BDSO obtained by both PEE (32.28±2.02%) and SC-CO2 (34.74±1.91%), which was consistent with the results of screw pressed seed oil (linolenic acid: 170.59±9.63 µg/mL) in the previous study [12]. These results indicate that the seed oil from B. dasystachya is a good source of linolenic acid.

Table 5. Fatty acid composition and content of B. dasystachya seed oil by PEE and SC-CO2 (area%, n=3)(Please see the attachment)

SFA

C11

C13

C14

C15

C16

C18

C20

∑SPA

PEE a

0.11±0.02

0.24±0.04

3.65±0.63

-

12.19±1.12

2.03±0.44

0.90±0.16

19.12

SC-CO2b

0.15±0.01

0.18±0.03

3.69±0.77

0.47±0.06

5.98±1.33

3.16±0.57

0.74±0.09

14.37

UFA

C16:1

C18:1

C18:2

C18:3

C22:1

∑UFA

∑PUFA

SFA/UFA

PEE a

0.63±0.14

23.08±1.58

21.41±1.09

32.28±2.02

3.47±0.25

80.87

53.69

0.2364

SC-CO2b

1.77±0.32

20.34±1.30

23.16±1.97

34.74±1.91

5.61±0.83

85.62

57.90

0.1678

a: BDSO obtained by petroleum ether extraction at 80 ℃ for 7 h;  b: BDSO obtained by supercritical CO2 fluid extraction at 25.00 MPa, 59.0℃, 2.25 SL/min. C11: Undecanoic acid, C13: Tridecanoic acid, C14: Myristic acid, C15: Pentadecanoic acid, C16: Palmitic acid, C18: Stearic acid, C20: Eicosanoic acid; C16:1: Palmitoleic acid, C18:1: Oleic acid, C18:2: Linoleic acid, C18:3: Linolenic acid, C22:1: Erucic Acid. SFA: saturated fatty acids, UFA: unsaturated fatty acids, PUFA: polyunsaturated fatty acids.

Comment 3: The authors should compare their results with previous reports such as Antioxidant and Anti-fatigue Activities of Seed Oil from the Berries of Three Indigenous Plants in Tibetan Plateau Lijuan Han, JingMeng, Yongjing Yang, Ying Ye, Yourui Suo, Journal of Food and Nutrition Research, 2015, Vol. 3, No. 7, 445-457. Response: Thanks a lot for the comments and recommendations. According to your reminder, we have reanalyze the results in manuscript, and we have highlighted with blue colored in our manuscript.(1) In line 53, in page 2, we summarized BDSO with our previous studies. “Previous studies have shown that the seed oil of B. dasystachya (BDSO) prepared by screw pressing has rich antioxidant contents, such as total polyphenol content (TPC, 281.61±19.06 g/g), total carotenoid content (TCC, 65.43±5.46 µg/g), total steroids content (TSC, 291.74±22.21 µg/g), unsaturated fatty acids (UFA, 368.61 µg/mL), and polyunsaturated fatty acids (PUFA, 305.39 µg/mL), some of which could reduce the amount of free radicals produced in vitro. Furthermore, our previous results showed that supplementation of BDSO could decrease the side effects of fatigue via variation of biochemical parameters and modify relevant antioxidant parameters in order to prevent lipid oxidation in forced swimming mice [12].”

  1. Han, L., Meng, J., Yang, Y., Ye, Y., Sou Y.R. Antioxidant and Anti-fatigue Activities of Seed Oil from the Berries of Three Indigenous Plants in Tibetan Plateau. J. Food Nutr. Res. 2015, 3: 445-457.

(2)In line 236, in page 9, we analyzed BDSO yield with our previous studies. “The yield of BDSO extracted by petroleum ether extraction (PEE) method was 10.76%. The BDSO yields obtained by SC-CO2 extraction at different conditions are present in Table 3. Previous studies [12] have shown that the yield of seed oil obtained by screw pressing was 5.32 ±1.72%, which was lower than the BDSO yield extracted by PEE and SC-CO2.”.

(3) In line 265, in page 9, we analyzed the infrared spectra of BDSO with our previous studies. “However, compared with the previous research results [12], the infrared spectra of the three kinds of seed oil (BDSO obtained by screw pressing, PEE, SC-CO2, respectively) have similar characteristic absorbance bands such as hydroxyl groups(around 3000 cm-1),  -CH2 (around 1460 cm-1), -CH3 (around 1377 cm-1), and - (CH2)n- (n≥4, around 720 cm-1).”

(4) In line 276, in page 10, we analyzed the fatty acid composition of BDSO with our previous studies. “In the previous study [12], the fatty acid content of BDSO obtained by screw pressing was analyzed by GC-MS, and the result was different from that of GC analyses in this paper: six saturated fatty acids (lauric acid, myristic acid, pentadecanoic acid, heptadecylic acid, stearic acid) and six unsaturated fatty acids (palmitoleic acid, oleic acid, linoleic acid, γ-linolenic acid, α-linolenic acid, eicosenoic acid) were identified. Furthermore, it was found that linolenic acid had the highest unsaturated fatty acid content in BDSO obtained by both PEE (32.28±2.02%) and SC-CO2 (34.74±1.91%), which was consistent with the results of screw pressed seed oil (linolenic acid: 170.59±9.63 µg/mL) in the previous study [12]. These results indicate that the seed oil from B. dasystachya is a good source of linolenic acid.”.

(5) In line 475 in page 18, we analyzed the antioxidant ability composition of BDSO with our previous studies. “Compared with previous study [12], the seed oils of B. dasystachya prepared by three methods (Screw pressing, PEE, and SC-CO2) exhibited prominent DPPH, ABTS radical scavenging and Fe3+ reducing activities in chemical-based assays. The activities of the seed oils showed a positive correlation to the individual concentrations. In comparison with BDSO obtained by PEE and screw pressing, highly concentrated (12 mg/mL) seed oils obtained by SC-CO2 could scavenge 89.56% and 92.60% of DPPH and ABTS radicals, respectively. The scavenging activity of the three seed oils (Screw pressing, PEE, and SC-CO2) upon addition of DPPH radical and the reducing power followed the order: SC-CO2 > PEE > Screw pressing. The scavenging activity of the three seed oils upon addition of ABTS radical followed the order: SC-CO2 > Screw pressing > PEE. Studies have revealed that the potential antioxidant function of plant oils is mainly because of polyunsaturated fatty acids (PUFAs), tocopherols, and other components [52]. Therefore, this sharp contrast might be due to the composition as well as the concentration of any anti-oxidant compounds present in the seed oils. At this point, BDSO extracted from SC-CO2 method could be considered as a valuable raw material for designing natural antioxidants.”

  1. Jiao, J., Li, Z.G., Gai, Q.Y., Li, X.J., Wei, F.Y ., Fu, Y .J., Ma, W. Microwave-assisted aqueous enzymatic extraction of oil from pumpkin seeds and evaluation of its physicochemical properties, fatty acid compositions and antioxidant activities. Food Chem. 2014, 147, 17–24.

Comment 4: English language and style are fine/minor spell check required.

Response: Thanks a lot for the comments and recommendation. We are very sorry for our incorrect writing. According to your reminder, we have corrected the mistakes in manuscript. We have revised the whole manuscript carefully and tried to avoid any error of the manuscript, we have highlighted with green colored in our manuscript. We very much appreciate your valuable suggestions and kind advice. We also noticed that we have made some mistake in the subtitles, we also revised it. We have rewritten the part of Statistical Analysis and Funding and highlighted with green colored words in manuscript as well. We highly value this opportunity and we have endeavored to revise the manuscript according to the instructions, comments, and suggestions from the editor and reviewers, necessary explanations, and corrections/revisions have been done carefully to improve the quality of the manuscript. The changes in the revised manuscript and supporting information have been highlighted in different color.

We are very grateful to you for your kind advice, helpful instructions and continuous effort in the processing of our manuscript and we will be very happy to provide any further information if needed.

Yours sincerely,Lijuan Han and Gang Li, DrQinghai UniversityXining, China, 810001Tel: +86-13678659123E-mail: [email protected];  [email protected]

Round 2

Reviewer 2 Report

Molecules (Manuscript ID: molecules-763959), Comments to the Authors:

Title: Characterization and biological activities of seed oil extracted from Berberis dasystachya Maxim. by supercritical carbon dioxide extraction method

Comments

After reading the authors response to my comments, I found the authors responded to all my remarks and I think the manuscript can be accepted for publication.